# Electrolyzed–Reduced Water: Review II: Safety Concerns and Effectiveness as a Source of Hydrogen Water

**DOI:** 10.3390/ijms232314508

**Published:** 2022-11-22

**Authors:** Tyler W. LeBaron, Randy Sharpe, Kinji Ohno

**Affiliations:** 1Centre of Experimental Medicine, Institute for Heart Research, Slovak Academy of Sciences, 841 04 Bratislava, Slovakia; 2Molecular Hydrogen Institute, Enoch, UT 84721, USA; 3Department of Kinesiology and Outdoor Recreation, Southern Utah University, Cedar City, UT 84720, USA; 4H2 Analytics, Henderson, NV 89052, USA; 5Division of Neurogenetics, Center for Neurological Diseases and Cancer, Nagoya University Graduate School of Medicine, Nagoya 466-8550, Japan

**Keywords:** electrolyzed reduced water, alkaline ionized water, high pH water, safety of ERW, water ionizer, molecular hydrogen, oxidation-reduction potential, electrolyzed oxidizing water

## Abstract

Many studies demonstrate the safety of alkaline-electrolyzed–reduced water (ERW); however, several animal studies have reported significant tissue damage and hyperkalemia after drinking ERW. The mechanism responsible for these results remains unknown but may be due to electrode degradation associated with the production of higher pH, in which platinum nanoparticles and other metals that have harmful effects may leach into the water. Clinical studies have reported that, when ERW exceeds pH 9.8, some people develop dangerous hyperkalemia. Accordingly, regulations on ERW mandate that the pH of ERW should not exceed 9.8. It is recommended that those with impaired kidney function refrain from using ERW without medical supervision. Other potential safety concerns include impaired growth, reduced mineral, vitamin, and nutrient absorption, harmful bacterial overgrowth, and damage to the mucosal lining causing excessive thirst. Since the concentration of H_2_ in ERW may be well below therapeutic levels, users are encouraged to frequently measure the H_2_ concentration with accurate methods, avoiding ORP or ORP-based H_2_ meters. Importantly, although, there have been many people that have used high-pH ERW without any issues, additional safety research on ERW is warranted, and ERW users should follow recommendations to not ingest ERW above 9.8 pH.

## 1. Introduction

Electrolyzed reduced water (ERW), also known as alkaline ionized water, is a popular type of “healthy water” worldwide [1]. ERW is produced through water electrolysis in which the cathode and anode electrodes are separated by a semi-permeable membrane. The water at the cathode produces ERW, whereas the water at the anode produces an acidic water called electrolyzed oxidizing water (EOW) [2]. The membrane prevents the two waters from mixing, allowing each to maintain their respective pH’s. These two types of water have a long history of use and have grown in popularity. Yet there remain many misconceptions and unknowns about them, specifically regarding ERW due to its many miraculous claims [3]. Some of these claims are substantiated by the literature and others are not. ERW contains hydrogen gas, which has recently been shown to exert many therapeutic effects [4]. This explains the long mystery of how/why ERW provided any biological benefit. However, despite the many favorable studies on ERW, there exist some important safety and health concerns. Some studies indicate that ERW can induce cellular injury and impair potassium homeostasis, among other risks. This article summarizes these safety concerns and discusses their potential mechanism(s). We first briefly summarize EOW and ERW benefits and history while also clarifying popular misconceptions about ERW. We then discuss the safety concerns of daily ingestion of ERW, followed by a discussion on the effectiveness of using ERW as a source of hydrogen water.

### 1.1. Electrolyzed Oxidizing Water (EOW)

The anodic water has an acidic pH and contains dissolved oxygen (O_2_ gas) and various chlorine species (e.g., Cl_2_, HOCl, OCl^−^). This is due to the oxidation of water [2H_2_O(*l*) → O_2_(*g*) + 4H^+^(*aq*) + 4e^−^] [5], and the oxidation of chloride ions (2Cl^−^ → Cl_2_ and Cl_2_ + H_2_O → HOCl + HCl), which occur at the anode. This type of water is often called EOW [6]. Due to the presence of chlorine species, this type of water is strongly oxidizing and is commonly used as a disinfectant and for sanitization. The effectiveness of its sanitization ability can be crudely estimated by measuring its oxidation-reduction potential (ORP). Its ORP measures around +600 mV to over +1200 mV and, if pH is controlled for, the ORP correlates with the concentration of chlorine species [7]. Thus, the effectiveness of EOW depends on the amount of chloride in the source water, the exposure time of electrolysis (for non-batch devices), the applied voltage to the electrodes, electrical conductivity of the water, and electrode configuration [6]. 

EOW at a slightly below neutral pH (≈4.5 to 6.5) is the most effective for sanitization due to the pH-dependent changes in chlorine species in which HOCl is predominant and is 100× more effective than OCl^−^, which predominates at alkaline pHs [8]. As the pH decreases below 3.0, the majority of the chlorine exists in the gaseous Cl_2_ phase, which is a less effective sanitizer than HOCl, and since it is volatile, poses a greater risk to health [9]. Therefore, the most effective EOW units are those where (i) sodium chloride can be added to ensure adequate electrical conductivity and the presence of chloride, (ii) the pH is maintained near neutral, (iii) and electrolysis can occur continuously for a set amount of time. These are typically batch-type electrolysis units with no membrane as opposed to flow-through machines, and therefore do not produce two separate types of water. In addition to ensuring reliable performance in producing the required HOCl concentration, this method also prevents the EOW from being too acidic and rendering it less effective and more toxic to human health. Generally, mesh electrodes are more effective compared to solid plate electrodes [10]. The chemistry and industrial uses of EOW have been extensively reviewed elsewhere [6,7].

### 1.2. Electrolyzed Reduced Water (ERW)

The other type of water, produced at the cathode, is ERW. ERW is produced at the cathode according to the reduction reaction [2H_2_O(*l*) + 2e^−^ → H_2_(*g*) + 2OH^−^(*aq*)]. This produces water having an alkaline pH due to the increased hydroxide ions (OH^−^) and a negative ORP due to the dissolved H_2_ gas. The pH of ERW may range from slightly alkaline to pH 11.5, the ORP may range from −300 mV to over −800 mV, and the dissolved H_2_ levels may range from less than 0.1 mg/L to near 1.6 mg/L. There have been numerous health claims made about the benefits of ERW, some supported by science and others that are either contradicted by scientific principles and/or refuted by scientific investigation. Research on ERW primarily started in the 1990s and it was found that ERW exerted biologically important benefits. For example, ERW has been shown to inhibit tumor growth [11], protect the liver from toxins [12], improve lipid metabolism [13], and provide other benefits as has been reviewed [1,14].

The benefits of ERW have been shown to be exclusively due to the presence of dissolved molecular hydrogen (refer to Part I of our tandem reviews for details) [3]. However, it was largely unknown that H_2_ gas exerted any therapeutic effect prior to 2007, when *Nat. Med* published a paper showing hydrogen’s promising therapeutic effects [15]. Since the therapeutic effects of H_2_ were recognized after the therapeutic effects of ERW were reported, the benefits of ERW were not initially attributed to molecular hydrogen [3]. Instead, the proposed agent in ERW included notions such as (i) the alkaline pH to neutralize toxic waste, (ii) the negative ORP to combat oxidative stress, (iii) altered water structure and “microclustering” to improve cellular hydration, (iv) abundance of hydroxide ions to neutralize free radicals, (v) active or atomic hydrogen, mineral hydrides, and even free electrons to scavenge free radicals [16,17]. These claims have been extensively reviewed and/or investigated and subsequently refuted. Scientifically, the only claim that appeared to have merit was the importance of the negative ORP [3].

#### ERW and Oxidation Reduction Potential

Studies found that when the ORP was no longer negative the therapeutic benefits were eliminated [3,18]. This became an important recognition, but the agent responsible for the negative ORP was not clear. Promoters of ERW claimed the negative ORP was ephemeral because it indicated things like a semi-stable “structured water”, electrically charged water, charged minerals, electrons, or the presence of negatively charged hydroxide ions [18,19,20,21]. Again, from a chemistry perspective, none of these claims were even logically feasible as they were refuted by basic physical and chemical principles. Nevertheless, despite not knowing what was responsible for the negative ORP, its importance in ERW continued to be confirmed by research and everyday consumers.

It was soon considered that the more negative the ORP the greater the benefits. Some studies even showed that a small negative ORP (e.g., −200 mV) did not provide any benefits. It is now well known that the negative ORP in ERW is due to the presence of dissolved H_2_ gas. Although it is true that the greater the hydrogen gas concentration, the more negative the ORP, the predominant influence on ORP is the pH (see Figure 1 in Part I for details). In alkaline water, only a very small level of H_2_ is needed to give a negative ORP. For example, a pH of 9 and an H_2_ concentration of 0.1 mg/L will measure an ORP of −497 mV. Increasing the H_2_ concentration 10x (0.1 to 1 mg/L), only decreases the ORP to −527 mV, which is within the margin of error of a typical ORP meter. Moreover, the same magnitude of change in the ORP occurs by increasing the pH by only ½ a pH unit (9 to 9.5). The reasons for this are based on the Nernst equation, a topic that has been extensively reviewed (see Figure 1 in Part I) [22]. Therefore, although the ORP can indicate the presence of H_2_ in ERW, it cannot be used to either estimate the concentration of H_2_ or even compare the level of H_2_ in one sample to another. In other words, since pH dominates the ORP reading, the level of H_2_ can easily be obscured. A higher pH with a low level of H_2_ will exhibit a more negative ORP, than neutral pH water with a higher level of H_2_.

However, it seemed that the greater the negative ORP the greater the therapeutic effects. This would likely be true for ERW because the methods used to make a greater negative ORP water are the same methods that would elevate the level of dissolved hydrogen gas. For example, a negative ORP could be made more negative by increasing minerals in the source water prior to electrolysis to improve its conductivity, elevating the applied voltage, increasing the electrode surface area, and slowing down the flow rate. Unfortunately, these changes also result in a very high pH (e.g., 10.5 to >11.5), which feed into the misconception that ingestion of high-pH alkaline water has therapeutic effects. Perhaps this gave rise to the current bottled alkaline water marketplace. Importantly, the benefits that were correlated with higher alkaline pH only occurred with ERW where there was also a negative ORP due to the higher hydrogen gas levels [22].

Although many researchers and consumers now recognize that the benefits of ERW are due to hydrogen gas, perhaps over 50% of the ERW studies still do not address its importance (See Figure 3 in part 1 of our tandem reviews). Moreover, while ERW can be a viable method to ingest hydrogen water, there are important safety concerns that should be addressed. Additionally, the effectiveness of ERW machines in making hydrogen water has important limitations that need to be considered.

## 2. Safety and Concerns of ERW Ingestion

For the most part, the safety of ERW has been well established since its initial approval by the Japanese Ministry of Health and Welfare in 1964 [2]. This has been confirmed by many cellular, animal, and human studies as detailed in previous articles [23,24,25]. This is particularly true for ERW’s therapeutic agent, molecular hydrogen, whose safety profile has been extensively reviewed previously [26]. However, ERW is not simply water containing dissolved H_2_, but it also has a high pH, and it potentially contains electrode particles that leach from the electrodes during electrolysis. Accordingly, there are some disconcerting studies and certain situations where additional caution should be considered for ERW [27,28]. Table 1 summarizes the most prevalent safety concerns with ERW. One of the most concerning comes from several animal studies in the 1990s that reported that rats drinking ERW developed marked necrosis and fibrosis of the cardiac tissue with subsequent hyperkalemia [29,30,31,32]. The exact reason and mechanism for these anomalous observations have not been elucidated. However, since it would not be due to the H_2_ gas, then it is most likely attributed to either the alkaline pH or more likely to electrode degradation, each of which is discussed below.

### 2.1. Safety of High pH Water

The WHO and EPA recommend that the pH of drinking water not exceed 8.5 [39]; however, as discussed earlier, alkaline water alone does not have significant buffering or acid-neutralizing capability, so it is unlikely to be a significant concern. However, if the pH of the alkaline water is excessively high, then the high pH itself can be problematic. Additionally, frequent ingestion of very-high-pH water may start to have meaningful buffering effects. Indeed, drinking a liter of 12.7 pH alkaline water is roughly equivalent to ingesting one tsp of baking soda. Long-term daily ingestion of one tsp of baking soda in a diabetic patient resulted in ventricular tachycardia and electrolyte abnormalities such as hypokalemia (low blood potassium) due to metabolic alkalosis [40]. Furthermore, rats given alkaline water at a pH 11.2–12 for one year resulted in significantly impaired growth and body weight. Some rats also presented with dull and patchy fur suggesting a systemic toxic or metabolic response [28]. Similarly, another study evaluated the effects of administering acidic (pH 3.4) or alkaline (pH 10.1) ionized water on embryonic development compared to neutral control water. It was revealed that both the acidic and alkaline groups resulted in a higher embryonic death, as well as impaired growth of the surviving fetuses [33].

Ingestion of water with a pH greater than 9.5 (e.g., pH 10–12) is not unlikely with many household alkaline water ionizers. Although discouraged by commercial companies, several alternative practitioners recommend drinking so-called “strong alkaline water” for heartburn, indigestion, food poisoning, stomach flu, arthritis, gout, muscle soreness and injuries, migraines, stroke, etc. [20,41]. This water is made using a saturated saline solution, which increases the conductivity of the water allowing the electrolyzer to produce a pH greater than 12. Additionally, many advocates of ERW promote drinking the “purple water”, referring to the dark purple color that results when testing the water using a common universal pH indicator. However, this purple color does not mean pH 9.5, but rather, as a sensitive indicator, the dark purple color does not appear until pH 10 or greater [42]. Therefore, if the consumer wishes to avoid drinking water above a pH 10, then they should not drink the water that will cause the universal pH indicator to turn dark purple.

However, even without ingesting “strong alkaline water”, and only setting the machine at 9.0 or 9.5, consumers may still be ingesting water above pH 10 on a daily basis. Unfortunately, the consumer may be completely unaware of this because, although most water ionizers allow the user to set the desired pH (e.g., 8.5, 9.0, and 9.5), these settings only change the applied voltage to the electrodes, where the higher pH setting has a greater applied voltage. They do not ensure that the pH produced is the pH indicated by the setting, as these units don’t have internal pH electrodes or feedback software, which arguably makes these types of buttons misleading. Therefore, if the source water has a high mineral concentration and/or is already slightly alkaline (e.g., pH > 8), then, at the “9.5 pH” setting, the pH of the produced alkaline water may easily exceed pH 10. Moreover, slowing the flow rate allows the water to undergo electrolysis for a longer time, which strongly influences the pH. Depending on the mineral concentration of the source water, slowing the flow rate to a “trickle” can result in alkaline water with a pH above 12. This is an impressive feat and doing so results in a more negative ORP due to the higher pH and H_2_ gas concentration (see Figure 1 in Part I); however, it also increases the risk of harmful effects.

Additionally, other harmful effects may include decreased nutrient absorption. For example, calcium precipitates at higher pHs, and an acidic pH facilitates the solubilization of minerals and the extraction of vitamins from foods. Ingestion of one liter of alkaline water at pH 10 can easily neutralize several hundred milliliters of pH-4 stomach acid. This becomes more serious if the body cannot easily respond because the individual lacks stomach acid (e.g., due to medications, impaired gastric acid production, surgical removal of the stomach, etc.). In these cases, the absorption of minerals, vitamins, and other important nutrients may be significantly impaired [36]. The neutralization of stomach acid may also increase the risk of a pathogenic bacterial infection that would have otherwise been killed by the acidic juices of the stomach [36]. Moreover, proton pump inhibitors change the gut microbiome and exacerbate small-intestinal mucosal injury in patients taking aspirin [43], which will be further discussed in Section 2.4. The gastric acid neutralization effects of high pH alkaline water may, in some cases, attenuate esophageal gastric reflux, while in other cases/people it exacerbates it.

### 2.2. Safety of Electrodes and Metal Degradation

Another potential concern is leaching of small amounts of platinum nanoparticles (PtNPs) during the electrolysis process. The safety of PtNPs has not been completely established [2] and there are potential toxicity concerns [34,35]. For example, PtNPs can interfere with ion channels in cardiomyocytes resulting in disruption of cardiac electrophysiology and life-threatening cardiac conduction block [44]. The toxicity of PtNPs appears to be size-dependent, with the smaller size inducing greater damage including hepatotoxicity and nephrotoxicity [45,46]. This makes PtNPs attractive for cancer treatment by inducing apoptosis via DNA fragmentation [47]. However, this also occurs in non-cancer cells in which PtNPs were incubated with human cells resulting in DNA damage and activation of p53 and subsequent apoptosis [48]. Moreover, Hiraoka et al. found that ingestion of water containing platinum led to an acute hepatic disturbance in healthy humans [49,50]. This is an ongoing field of research that shows potential benefits in certain cases such as cancer, and toxic effects in others [51].

However, as previously mentioned, the amount of PtNPs is generally below the detection limit, and PtNPs seem to only appear when doing continuous electrolysis. Nevertheless, this concern increases when the source water contains more minerals, when the applied voltage increases (e.g., highest alkaline setting), and when the flow rate is slowed down allowing a longer electrolysis time and elevated electrode temperature; these conditions facilitate electrode degradation. Moreover, these electrodes are not pure platinum, but platinum-coated titanium. These metals themselves may not be of the highest purity, and may easily contain traces of other heavy metals known to be toxic including nickel, cadmium, lead, cobalt, arsenic, etc. Loss of platinum coating and/or degradation of the electrode can result in these toxic metals accumulating in the alkaline drinking water. Similarly, chemical degradation of the membrane that is used to separate the acid and alkaline chambers may also be problematic [52]. The consequences of electrode/membrane degradation may be responsible for the marked necrosis and fibrosis of the cardiac tissue and subsequent hyperkalemia in rats drinking ERW [29,30,31,32].

### 2.3. Hyperkalemia

As in the aforementioned animal studies, hyperkalemia is also the most concerning report observed in some human clinical studies. Specifically, in some people with poor kidney function, drinking ERW above a pH of 10 can result in hyperkalemia [37,38]. However, the mechanism responsible for this elevation in blood potassium levels remains unclear. Importantly, hyperkalemia from ingesting alkaline water is somewhat counterintuitive because, even if alkaline water could substantially increase the blood pH, then blood potassium levels would decrease, not increase. This would be due to the K^+^/H^+^ antiporter, which would work to lower the pH back to normal by moving H^+^ out of the cell in exchange for K^+^ that would go into the cell.

The exact reason for hyperkalemia is unknown, but there are at least two potential explanations. One potential, although unlikely, explanation is that high pH ERW above pH 10 can significantly increase the HCl production [38]. The physiological mechanism that creates HCl relies on the parietal cell H^+^/K^+^ ATPase. The increased secretion of H^+^ ions into the gastric lumen results in an increase in intracellular K^+^ ions in the parietal cell. These potassium ions then get excreted out of the cell into the extracellular space via basolateral Na^+^/K^+^ ATPase pumps. Similarly, alkalization of the intestinal lumen may initially increase the enterocytes’ intracellular K^+^ concentration via the K^+^/H^+^ antiporter. The cell responds by activating basolateral Na^+^/K^+^ ATPase pumps, resulting in normal intracellular K^+^ levels, but higher extracellular K^+^ levels. If this occurs in individuals with poor kidney function, they may be unable to maintain normal serum potassium levels, thus resulting in the observed hyperkalemia.

The other potential reason may be cytotoxic damage to cells as was the case in the earlier animal studies [29,30,31,32]. In these studies, ERW induced cell damage and myocardial lesions, which then released K^+^ into the blood resulting in hyperkalemia. The active agent in ERW responsible for these toxic effects remains unknown, but the most probable culprit is toxic metals leached from the electrodes. Degradation of the electrodes occurs mostly when making the higher pH ERW, due to the higher voltage and/or longer electrolysis time, greater heat generation, and a higher mineral concentration in the source water. This correlates to the observation that hyperkalemia occurs only when the pH exceeds 10. Therefore, ERW with a pH of 10 or greater may contain more cytotoxic metals, which damage the cells. The damaged cells then release potassium into the serum, and, if the individual is unable to maintain normal serum potassium levels due to poor kidney function, then hyperkalemia occurs.

The prevalence of hyperkalemia is well recognized, which is why Japanese and Korean regulations on these alkaline water ionizers mandate that the pH of the alkaline drinking water should not exceed pH 9.8 [37]. Similarly, the official owner’s manual of Enagic states: “*when drinking KANGEN water, adjust the pH value to 9.5 or lower. It is not recommended to drink water with a pH over 10.0. Check the pH regularly*.” [53]. It also states: “*Do not drink KANGEN Water if you have kidney problems such as kidney failure or trouble processing potassium*.” [53]. These safety recommendations appear prudent to minimize potential concerns with ingesting high pH water, hyperkalemia, and decrease exposure to metal electrode particles. However, despite the general safety of ERW for most people following these recommendations, further investigation of these possible and observed health concerns with ERW is warranted.

### 2.4. Safety of Gut Microbiota

The effects of ERW on gut microbiota have been reported in mice [54] and humans [55]. All these studies used hydrogen-enriched ERW. The shared feature was that ERW increased short-chain fatty acid (SCFA)-producing bacteria and fecal concentrations of SCFA. SCFA enhances the expression of the Foxp3 gene and induces the differentiation of naive T cells into regulatory T cells by inhibiting histone deacetylases, which should have an anti-inflammatory effect [56]. Inhibition of histone deacetylase also thickens intestinal mucosa, which is partly due to the role of SCFA as an energy source for intestinal epithelial cells [57]. Long-term administration of proton pump inhibitors (PPIs) largely changes gut microbiota, probably by increasing gastric and duodenal pH in rats [58] and humans [43,59,60,61]. In humans, an increased relative abundance of *Streptococcus* [43,59,60,62] is a shared feature of PPI administration. A relative abundance of intestinal *Streptococcus* is increased in patients with atopic dermatitis and is negatively correlated with SCFAs [63]. A relative abundance of intestinal *Streptococcus* is also increased in coronary atherosclerosis [64]. In addition, in humans, long-term administration of PPIs significantly decreases a relative abundance of *Faecalibacterium*, which is a major SCFA-producing bacterium in our intestines [61]. Although *Streptococcus* was not increased in any reports on the effects of ERW on gut microbiota, and SCFA-producing bacteria were rather increased by ERW [54,55,65], alkalinity of high pH ERW may exert unfavorable effects on gut microbiota.

## 3. Effectiveness of ERW Machines for Making H_2_ Water

For many decades ERW was the primary method of obtaining hydrogen water. Logically, this is because it was not yet known that H_2_ was the important agent responsible for the therapeutic benefits of ERW. Water ionizer manufacturers focused their attention on changing the pH and were not concerned about the concentration of molecular hydrogen. In fact, in the early days of alkaline water ionizers, before the benefits or high safety of H_2_ were known, some manufacturers may have worked to develop machines that produced high alkaline pH with low dissolved H_2_. For whatever reason, not all conventional alkaline water ionizers are equal in their ability to provide dissolved hydrogen water. Most alkaline water ionizers, when brand-new with optimal source water and normal flow rate, make a concentration of dissolved H_2_ in the range of 0.2 mg/L to 1.2 mg/L within the pH range of 8.5 to 10.5. The concentration depends on the machine, electrode material, surface area and morphology, applied voltage, the mineral content of the source water, the flow rate, and the cleanliness of the electrodes, all of which may result in an H_2_ range of less than 0.01 mg/L to nearly 2 mg/L over the pH range of ≈8 to over 12.

Slowing the flow rate allows the water to remain in contact with the electrodes during electrolysis for a longer period of time and thus more production of H_2_, as well as OH^−^ ions out of a smaller volume of water, which increases the pH. If there are enough minerals in the water, then slowing the flow rate down to get a 1 mg/L H_2_ concentration often results in a pH greater than 10. Although this may be a sufficient concentration of H_2_, it may also be problematic for the pH to be so high due to the concerns mentioned above. In fact, based purely on stoichiometry without considering the weak buffering effects of the original source water or proton leakage into the cathode, the production of a 1 mg/L H_2_, requires one millimole of H^+^ and produces one millimole of OH^−^ ions. This results in a pH of 11. Similarly, again solely based on pure stoichiometry, the production of only a 0.1 mg/L of H_2_ results in a pH of 10, which makes sense because pH is logarithmic. However, not all the H_2_ produced gets dissolved into the water. This is easily observed by the sometimes foggy or cloudy appearance of the water, which indicates undissolved gas. Similarly, a flame can be held underneath the running water, which ignites the H_2_ gas resulting in a crackling/popping sound. This frequently-used marketing demonstration attempts to “prove” the high levels of H_2_ dissolved in the water, but unfortunately, it actually demonstrates the presence of undissolved H_2_ gas, since fully dissolved H_2_ gas, like gunpowder when dissolved in water, cannot be ignited. Therefore, even if enough H_2_ gas is produced to make a 1 mg/L of H_2_ water, not all of the H_2_ gas dissolves in the water. The H_2_ that is not dissolved in the water can’t provide benefits since it is quickly lost into the atmosphere. The estimated dissolution ratio for water ionizers is probably around 25%, but ranges from < 1% to maybe 70% at the high end. This means that, if the electrodes are clean and the mineral concentration is optimal, then, at the highest dissolution ratio, only a 0.07 mg/L H_2_ concentration can be made before the pH increases above 10. In practice, however, depending on the ionizer, we can sometimes measure closer to 0.7 mg/L H_2_ before the pH exceeds 10. Nevertheless, ingesting ERW with a pH above 9.8 is not recommended [37].

It is estimated that, in order for the alkaline ionizer to produce significant changes in pH and H_2_ concentration, the source water must have a minimum mineral concentration of around 50 mg·L^−1^ [2]. In an attempt to overcome this issue, some machines allow you to inject additional minerals (e.g., calcium glycerophosphate) into the source water prior to electrolysis, and/or use filters containing calcium sulfite that may marginally increase the mineral level. Unfortunately, these methods neither significantly increase the mineral concentration nor do they provide an effective long-term solution. This means that customers purchasing water ionizer devices who live in areas with low mineral concentrations will not get any meaningful levels of molecular hydrogen in their water. However, if there are enough minerals in the water, then this creates two other concerns, (i) excessively high pH, which is associated with hyperkalemia and electrode degradation along with its associated concerns, and (ii) required frequent cleaning of the machine. Minerals, specifically calcium ions, which are positively charged, tend to accumulate on the negatively charged cathode resulting in limescale build-up. These calcium deposits not only decrease the effective surface area of the electrode but importantly also prevent the H_2_ gas from dissolving in the water. The authors have observed that, within as little as two weeks, some alkaline water ionizers may go from producing nearly 1 mg/L to less than the detected level of 0.01 mg/L. However, after cleaning the machine with citric acid, the H_2_ concentration returns to normal levels. This observation is not widely known because of (i) insufficient awareness of the importance of H_2_ in ERW, (ii) the overt focus on pH, microclustering, or other unimportant or unfalsifiable properties, and (iii) the use of ORP meters, which still give a very negative ORP despite very low levels of H_2_ due to the very high pH, (e.g., pH, 10.5, ORP = −580 to −730 mV, yet H_2_ < 0.1 mg/L) [22].

These observations are illustrated in Figure 1, which shows the risk-to-benefit ratio of ERW. As the H_2_ concentration increases, the pH also increases along with the associated risks mentioned in Table 1 (e.g., metal toxicity, tissue damage, hyperkalemia, impaired nutrient absorption, etc.). The ERW range is depicted in blue where most of it is in areas that are not recommended (zones 1, 3, and 4). This is because either the risk is low, but so is the H_2_ concentration (zone 3), or the risks are high for both the high H_2_ (zone 1) and low H_2_ (zone 4) areas. Only a small portion of the ERW range enters the recommended area (zone 2). This is where the H_2_ concentration is at least slightly above the potential therapeutic threshold, while the risks are at a minimum (e.g., pH below 9.8). Unfortunately, this optimal, albeit small area may not be possible for all ERW devices. In fact, as mentioned, even an ERW device that can otherwise produce sufficient levels of H_2_ at a pH lower than 9.8, may quickly develop scale on the electrodes, which would result in more of a horizontal line with a therapeutically ineffective H_2_ concentration and has a high potential for risk.

### Other Methods of Providing and Administering Molecular Hydrogen

In addition to obtaining molecular hydrogen through drinking water containing the dissolved gas, there are also many other methods of providing molecular hydrogen [66]. These include inhalation, intravenous H_2_-rich saline, H_2_-saline eye drops, H_2_ hyperbaric, H_2_ bath, H_2_-producing ingestible capsules, and novel H_2_ donors [67]. However, hydrogen water remains the most common method of H_2_ administration [68].

Now that molecular hydrogen is recognized as being the therapeutic agent in ERW, many manufacturers/companies have developed products focused only on providing high levels of molecular hydrogen in the water [66]. These include neutral-pH electrolyzed reduced water, neutral-pH H_2_-infusion machines and portable electrolysis devices that use proton exchange membranes, high-concentration H_2_-producing tablets [69], and even ready-to-drink H_2_-infused functional beverages [70,71]. These other methods completely circumvent the unique requirements of both ERW machines and the source water to make clinically relevant levels of dissolved H_2_. Moreover, at least the non-electrolytic methods also circumvent the safety concerns of ERW listed in Table 1. The neutral-pH electrolytic devices likewise circumvent most if not all these concerns due to their lower pH. However, since metal electrodes are still used, one cannot rule out the possibility of metal contamination in the drinking water. Nevertheless, this risk may also be mitigated by the neutral-pH machines because (i) neutral pH is not as corrosive on the electrodes, whereas basic conditions accelerate degradation of platinum electrodes [72], (ii) a lower current density results in less electromigration and mechanical stress and less electrode degradation [73], (iii) neutral-pH devices produce lower heat on the electrodes, which also decreases the debonding of the platinum [73], and (iv) some certain metal impurities are more prone to leaching and/or more soluble and thus more bioavailable at higher pHs (e.g., platinum, lead, chromium, arsenic, aluminum, etc.) [74,75].

## 4. Recommendations and Guidelines

If ERW is to be used for hydrogen water, the investigator and consumer need to ensure that the mineral concentration of the source water is optimal, the electrodes are free from calcium deposits, the flow rate is optimal to prevent the pH from exceeding a pH of 10, and finally that they measure the H_2_ concentration to ensure that a therapeutic and consistent level of H_2_ is achieved.

For those who choose to use ERW machines as a method for making either EOW or H_2_ water, the following guidelines and recommendations are provided:

For using EOW
The most effective pH for EOW is between 4.5 and 6.5.The HOCl concentration depends on source water chloride concentration, flow rate, applied voltage, electrode configuration, and plate morphology, which may all have wide variations.With all other factors being equal, mesh electrodes are more effective at producing HOCl than solid plate electrodesIf the pH is within the optimal range, the ORP can be used to indicate the sanitizing/disinfecting ability and should generally exceed +700 mV to over +1200 mVUsing chlorine test strips can also be helpful to ensure the HOCl concentration is high enough.

For drinking ERW
Ensure that source water has a mineral concentration (total dissolved solids, TDS) sufficient to permit electrolysis to occur (minimum TDS varies with machine design)If source water does have sufficient minerals, be sure to clean the machine as needed.○Although some machines have self-cleaning modes that reverse the electrode polarity, this may not be sufficient, and will not prevent scale build-up in other parts of the machine including hoses and solenoid valves.○Cleaning the machine with weak acids such as vinegar or citric acid can dissolve the calcium and mineral salts and remove them from the electrodes and other wetted components.Frequently measure the H_2_ concentration of ERW to ensure the level is within the desired range. Be sure to use accurate methods (e.g., gas chromatography, or perhaps the less accurate but useful redox titration reagent (MiZ Japan and H_2_Blue™, USA) (see also [22]) to measure H_2_. Specifically, avoid using pH-sensitive methods such as ORP and ORP-based H_2_ meters (see [22]).○A low H_2_ concentration may indicate insufficient minerals in the source water, a flow rate that is too high, or a need to clean the machine with an acidic solution.Ensure that the ERW unit is from a high-quality supplier, the electrodes are coated with high purity-platinum, and the thickness is great enough to withstand normal operation.Follow governmental regulations to not ingest ERW that exceeds pH 9.8 to prevent hyperkalemia and other issues associated with ingesting high pH ERW.○Remember, even if the setting/button states the pH, this simply alters the applied voltage, thus the actual pH may be higher or lower.○If a universal pH indicator is used, the color should not be “dark purple” as this indicates a pH greater than 10.

## 5. Conclusions

ERW has been consumed for over half a century and has been subjected to many studies and claims. There are many studies confirming the therapeutic effects of ERW that predate and post-date the research on molecular hydrogen. However, the benefits of ERW have been clearly shown to be due to the dissolved molecular hydrogen and not the other claimed properties. The negative ORP in ERW is an important indicator of whether ERW will provide any benefits only because it indicates the presence of dissolved hydrogen gas. However, since ORP meters and ORP-based H_2_ meters cannot be used to accurately measure the concentration of H_2_ in ERW, their use is discouraged. Despite the therapeutic effects of ERW and its relatively high safety profile, there still exist important safety concerns with ERW, especially at high pH. The most concerning are the reports of hyperkalemia, which most likely may be attributed to tissue damage induced by metals leached into the water during electrolysis. However, under the right conditions, the high pH of ERW alone may also induce harmful effects in those who are more susceptible to its effects. It is recommended that users of ERW do not ingest ERW when the pH is greater than 9.8 in order to stay within the guidelines of company statements and governmental regulations. Due to the potential safety concerns of ERW, some users may seek alternative methods capable of providing more consistent and higher H_2_ concentrations without requiring as much maintenance and frequent measuring of H_2_. Nevertheless, despite the relatively low levels of H_2_ provided by ERW when the pH does not exceed 10, ERW remains a common and simple method of providing hydrogen water. If this method is chosen by the researcher or consumer, it is important that the level of H_2_ is frequently monitored with an accurate method since the concentration of H_2_ may unknowingly be well below the therapeutic threshold.

## Figures and Tables

**Figure 1 ijms-23-14508-f001:**
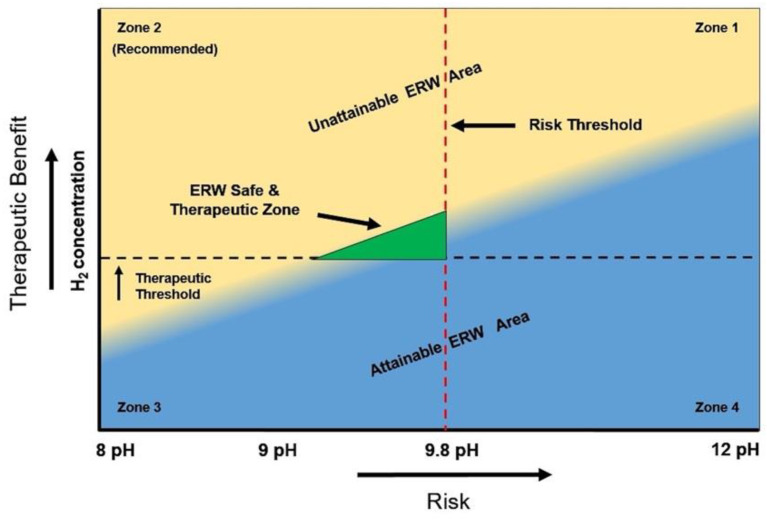
Representation of the risk-to-benefit ratio of ERW in making dissolved hydrogen gas and potential health risks. The ERW range only overlaps a small portion of the recommended zone 2 in which the H_2_ concentration is sufficient, and the risks are minimized.

**Table 1 ijms-23-14508-t001:** Potential harmful effects from ingesting high pH alkaline ionized water.

Potentially Harmful Effects	Comment	References
Growth retardation	Only animal studies and drinking very high pH constantly	[28,33]
Tissue necrosis and damage	Only animal studies. Could be due to electrode degradation	[29,30,31,32]
Electrode toxicity platinum nanoparticles (PtNPs) and other heavy metals	Depends on electrode purity, voltage, duration, etc.	[32,34,35]
Accumulation of harmful substances in drinking water	If harmful substances were not removed, some substances may accumulate in the alkaline drinking water	[33]
Impaired mineral, vitamin, and nutrient absorption	Unlikely for a healthy individual, but easy to induce with ingestion of high-volume high pH water	[36]
harmful bacterial overgrowth	Bacteria do not survive for long under pH 4. Neutralization of stomach acid could be problematic	[36]
Hyperkalemia	Observed in animal and human studies if pH exceeds 10. Cause remains unknown	[32,37,38]
Excessive thirst	Commonly reported when pH exceeds 10. Animals also drink more water. May be due to irritation of the mucosal lining	[3,28]
Exacerbation of gastric reflux	Commonly reported in some individuals, whereas in others it seems to help.	Anecdotally reported.

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
