# Peer review of "Electrolyzed–Reduced Water: Review II: Safety Concerns and Effectiveness as a Source of Hydrogen Water"

_ijms, 2022, doi:10.3390/ijms232314508_

Round 1
Reviewer 1 Report
Reviewer's Comments:
The authors have performed a detailed analysis of alkaline electrolyzed reduced water (ERW) in terms of its safety and efficacy in this review. As a result, the authors showed that hyperkalemia has been reported in experimental animals and humans, and significant tissue damage has also been observed in experimental animals. The authors also showed that these safety issues are due to the platinum nanoparticles leached into the ERW, other metals with harmful effects, and the high pH of the ERW. The authors also showed that it is important for ERW users not to consume ERW with a pH above 9.8, since a pH above 9.8 can cause safety issues.
I believe that the analysis in this review should be adopted, because it is very accurate and provides useful information and suggestions not only to the readers of the International Journal of Molecular Science but also to manufacturers and distributors of ERW. However, the following a minor revision is needed.
Since I too believe, as the authors claim, that molecular hydrogen is the main body of the efficacy of ERW, the problem of high pH of ERW can be solved by using neutral electrolyzed reduced water. Please explain the advantages of making the electrolyzed water alkaline rather than neutral.

Author Response
We appreciate your kind comments and understanding of the importance of this article. We also appreciate your suggestion regarding the advantage of neutral-pH ERW. Although we previously included neutral-pH hydrogen water, we have significantly expanded on this area in section 3.1 (lines 430 to 446). We hope that our addition/modification addresses your valuable comment.

Reviewer 2 Report
Dear Editor,
The manuscript "Electrolyzed Reduced Water: II. Safety Concerns and Effectiveness as a Source of Hydrogen Water" discusses the safety concerns of electrolyzed reduced water (ERW) and the recommendations for choosing the best ERW device and its effective and healthy measures of use. The manuscript was well constructed with sound language. However, there are minor comments that I have included in the pdf file in the note boxes.

Author Response
We appreciate your positive comments on our article. Also thank you for your suggested changes. We made some minor grammatical improvements to the English as highlighted in yellow in the revised manuscript.
Comment 2-1
“Count” should be added to the bacterial names.
Answer 2-1
We appreciate your valuable suggestion. We added “the relative abundance of” for each bacterium. Absolute bacterial counts were scarcely reported in the past, but not in these days.
Comment 2-2
Please add the first and earliest work performed on H2-fortified beverage:
Alwazeer, D., Delbeau, C., Divies, C., & Cachon, R. (2003). Use of redox potential modification by gas improves microbial quality, color retention, and ascorbic acid stability of pasteurized orange juice. International journal of food microbiology, 89(1), 21-29.
Answer 2-2
Thank you for the suggested early reference. We have included this into our manuscript as reference 71.
Comment 2-3
“Within as little as two weeks, some alkaline water ionizers may go from producing nearly 1 mg/L to less than the detected level of 0.01 mg/L.” Which model was this?
Answer 2-3
We appreciate your comment. However, we hope that you understand that our manuscript is not for criticizing any specific brand. We hope that the reader of the article will simply conclude that if they are using an ERW device, then this is a potential problem that can occur, which means they should follow our recommendations and frequently test the concentration of H2.
